# Simultaneous Silencing of Gut Nucleases and a Vital Target Gene by Adult dsRNA Feeding Enhances RNAi Efficiency and Mortality in *Ceratitis capitata*

**DOI:** 10.3390/insects15090717

**Published:** 2024-09-19

**Authors:** Gennaro Volpe, Sarah Maria Mazzucchiello, Noemi Rosati, Francesca Lucibelli, Marianna Varone, Dora Baccaro, Ilaria Mattei, Ilaria Di Lelio, Andrea Becchimanzi, Ennio Giordano, Marco Salvemini, Serena Aceto, Francesco Pennacchio, Giuseppe Saccone

**Affiliations:** 1Department of Biology, University of Naples “Federico II”, 80126 Naples, Italy; sarahmaria.mazzucchiello@unina.it (S.M.M.); francesca.lucibelli@unina.it (F.L.); marianna.varone@unina.it (M.V.); dora.baccaro@gmail.com (D.B.); matteiilaria678@gmail.com (I.M.); egiordan@unina.it (E.G.); marco.salvemini@unina.it (M.S.); serena.aceto@unina.it (S.A.); 2Department of Agricultural Sciences, University of Naples “Federico II”, Portici, 80055 Naples, Italy; noemi.rosati@unina.it (N.R.); ilaria.dilelio@unina.it (I.D.L.); andrea.becchimanzi@unina.it (A.B.); f.pennacchio@unina.it (F.P.)

**Keywords:** *Ceratitis capitata*, pest insects, RNAi-based biopesticide, vital gene, dsRNA nucleases, mortality rates, dsRNA degradation

## Abstract

**Simple Summary:**

The control of insect pest species, mainly belonging to Orthoptera, Hemiptera, and Coleoptera orders, can be based on novel emerging species-specific pesticides. These consist of dsRNA molecules delivered by feeding to insect larvae or adults, which suppress vital gene functions by RNA-RNA sequence complementarity and RNA interference. However, fewer studies of dsRNA feeding have been performed in dipteran pest insects. Two studies in Orthoptera and Coleoptera species have shown that suppressing intestinal enzymes degrading exogenous dsRNA can improve insect mortality rates. *Ceratitis capitata* (Tephritidae), the Mediterranean fruit fly (*Medfly*), is a major dipteran pest significantly impacting fruit and vegetable farming. Currently, its control heavily relies mainly on chemical insecticides, which pose health risks and have effects on beneficial pollinators. Previous attempts to induce mortality by adult dsRNA feeding in this and other Tephritidae species, such as *Bactrocera tryoni* and *B. dorsalis*, showed some effectiveness, but were often limited. We improved this method by simultaneously silencing two intestinal nucleases and a vital gene. We found a mix of three dsRNAs able to induce much higher mortality (79%) within one week, following only three days of adult feeding.

**Abstract:**

*Ceratitis capitata*, known as the Mediterranean fruit fly (*Medfly*), is a major dipteran pest significantly impacting fruit and vegetable farming. Currently, its control heavily relies mainly on chemical insecticides, which pose health risks and have effects on pollinators. A more sustainable and species-specific alternative strategy may be based on double-stranded RNA (dsRNA) delivery through feeding to disrupt essential functions in pest insects, which is poorly reported in dipteran species. Previous reports in Orthoptera and Coleoptera species suggested that dsRNA degradation by specific nucleases in the intestinal lumen is among the major obstacles to feeding-mediated RNAi in insects. In our study, we experimented with three-day adult feeding using a combination of dsRNA molecules that target the expression of the ATPase vital gene and two intestinal dsRNA nucleases. These dsRNA molecules were recently tested separately in two Tephritidae species, showing limited effectiveness. In contrast, by simultaneously feeding dsRNA against the *CcVha68-1*, *CcdsRNase1*, and *CcdsRNase2* genes, we observed 79% mortality over seven days, which was associated with a decrease in mRNA levels of the three targeted genes. As expected, we also observed a reduction in dsRNA degradation following RNAi against nucleases. This research illustrates the potential of utilizing molecules as pesticides to achieve mortality rates in *Medfly* adults by targeting crucial genes and intestinal nucleases. Furthermore, it underscores the importance of exploring RNAi-based approaches for pest management.

## 1. Introduction

There are 4000 species of fruit flies worldwide, and 35% are important pests, including commercial fruits with high economic value [1]. The Mediterranean fruit fly (*Medfly*), *Ceratitis capitata* (Diptera: Tephritidae), is a major agricultural invasive pest that inflicts widespread damage on various fruit crops globally [1]. The females lay eggs inside ripe fruits, and the subsequent hatched larvae feed them, damaging them. It causes yearly losses of billions of euros due to decreased production, increased costs of control methods, reduced marketability of affected produce, and lost markets [2]. Traditional control methods, like chemical insecticides, face several challenges, including environmental impact (such as pollinator reduction and pollution), resistance development, and diminished effectiveness [3]. Consequently, innovative and sustainable solutions are urgently required to significantly reduce the use of chemical pesticides in agriculture [4]. Biological control methods have been explored as a more environmentally friendly alternative to traditional insecticides for managing fruit fly populations. These methods include suppressing fruit fly populations using natural enemies, such as parasites and predators [5]. However, despite efforts in biological control, fruit fly control programs still face several challenges [1]. The Sterile Insect Technique (SIT) presents a potentially effective, eco-friendly, and species-specific control method [6]. However, its implementation is costly and challenging, mainly due to the need for extensive area-wide coordination in the suppression plan. Therefore, novel and sustainable approaches are urgently needed in parallel to or in combination with the SIT and other eco-friendly approaches.

One promising approach for pest control is using RNA interference (RNAi) to target essential genes by feeding [7,8]. Insect pests can consume artificial dsRNA molecules incorporated into their food or bait, causing a reduction in the target protein production that can affect the biological processes critical for their survival, growth, or reproduction, leading to phenotypic changes or even mortality. Over 15 years ago, Baum et al. [9] provided the first proof that RNAi could target specific beetle pests. Since then, numerous studies have demonstrated the effectiveness of this control strategy across various pest insect species. Environmental RNA interference (eRNAi) is emerging as a powerful tool for gene silencing, with significant potential for developing species-specific and environmentally friendly pest control methods [10,11]. The first RNAi commercial spray pesticide (Ledprona, GreenLight Biosciences, Durham, NC) was authorized by the U.S. Environmental Protection Agency (EPA) this year, designed as dsRNA to target after feeding the vital gene encoding the proteasome subunit beta of the *Colorado potato beetle* (CPB, *Leptinotarsa decemlineata* [12]; patent WO 2020/097414).

Among potential dsRNA-based pesticides, another relevant vital target is the *vacuolar (H+)-ATPase* (*v-ATPase*) complex, composed of two domains (transmembrane V0 and cytoplasmic V1 domains) and responsible for transporting protons (V0) across membranes using energy from the hydrolysis of ATP (V1) [13]. V0 and V1 domains are composed, respectively, of five (a–e) and eight subunits (A–H). The *v-ATPase* complex also includes two additional accessory proteins (AC45 and M8.9). Single genes encode some subunits, while most are by multiple genes. For example, in the genome of *Drosophila melanogaster*, 33 genes encode the 15 subunits of the *V-ATPase* [14]. *D. melanogaster vacuolar ATPase* loss-of-function mutations cause a lethal phenotype [14,15].

Due to its wide evolutionary conservation, *V-ATPase* is a promising target for pest control development. The *vacuolar ATPase* complex is also present in the gut epithelial cells and Malpighian tubules [16], and it has demonstrated susceptibility to RNAi-based gene knockdown when orally administered in various insect species. Some species of insect pests feed on the plants’ external portions, and they can be targeted by environmental dsRNA during this life stage. Feeding larvae of major pests, including Lepidoptera, Coleoptera, and Hemiptera, with *v-ATPase*-specific dsRNAs (targeting one of the various subunits, such as V-ATPase A, B, D, E, or H) led to significant mortality, ranging from 40% up to 100% (see, for example, [9,17,18,19]). In the Tephritidae *Anastrepha fraterculus*, soaking larvae (30 min) in dsRNA solution (500 ng/µL) targeting the chaperone protein (homolog of the human VMA21 integral membrane protein), required for proper v-ATPase assembly, led over the course of one week to a 25% increase in mortality compared to the 15% mortality rate induced by the dsGFP control [20]. However, insect pests of the Tephritidae family grow as larvae inside the fruit crop. Only the development of transgenic plants expressing dsRNA can solve the delivery problem at this life stage [21]. Nevertheless, adult dsRNA feeding can also be an effective control strategy, as shown in Hemiptera, Coleoptera, and Diptera, sometimes leading to strong mortality [22,23,24,25]. For example, targeting, by RNAi, the *vacuolar ATPase A subunit* (*BtvATPaseA*) in the sap-sucking pest *Bemisia tabaci* (Hemiptera), also known as white fly, caused up to 97% mortality after six days of feeding [26] (Table 1).

However, as the ingested dsRNA molecules are exposed to various gut enzymes, their stability and ability to reach the target cells without degradation affect the efficacy of gene silencing [35,36,37,38]. Insect dsRNA intestinal RNases seem to be naturally involved in the innate immune response against invading nucleic acids such as RNA viruses, and they can heavily limit the efficacy of orally delivered dsRNAs in inducing mortality [39,40,41]. Hence, research is required to explore and optimize dsRNAs’ stability in pest management applications [31,42]. Various strategies can address this issue.

Interestingly, the efficacy of the oral delivery of dsRNAs is improved when combined with the silencing of intestinal dsRNA nucleases of various pest insect species [32,43]. Tayler et al. [33] identified two gut dsRNA nucleases in the Australian Tephritidae *B. tryoni* and targeted them by oral RNAi together with a “reporter” gene, *yellow*. They found that co-feeding adults with three dsRNAs led to a significant reduction in *yellow* mRNA levels and enzymatic function, compared with feeding *yellow* dsRNAs alone. Similarly, Ahmad et al. [34] demonstrated in the larvae of *Zeugodacus cucurbitae* (Diptera) that simultaneous silencing of the intestinal nuclease *ZcdsRNase1* and the vital target gene *ZcCOPI-alpha* induced 84% mortality and a significant improvement in RNAi efficacy (Table 1).

Our study explored the effects of co-feeding with dsRNAs silencing two intestinal nucleases and the vital gene *vATPaseA* subunit. In the *Medfly* genome, we identified a *B. tabaci v-ATPase A* orthologue and two *B. tryoni dsRNAses* orthologues [44]. We observed that adult oral dsRNA co-feeding for three days with the three dsRNAs induced a substantial reduction in target RNAs and, after six days, 79% mortality. Furthermore, we found that in the gut juice of *C. capitata* flies fed with dsRNA silencing of intestinal nucleases, the degradation of target dsRNAs in vitro is less efficient, as previously shown in *B. tryoni* by Tayler et al. [33]. Compared with previous studies performed in many different insect species, we achieved significantly high mortality, opening the road to developing an effective formulation of dsRNA biopesticide for *C. capitata* (Table 1).

## 2. Materials and Methods

### 2.1. Insect Rearing

The *C. capitata* strain was reared in laboratory conditions at a specific temperature (25 °C) and humidity (70%) with a light–dark cycle of 12:12 h. Adult flies were fed artificial sugar and a powdered yeast extract diet in a 3:1 ratio, and with water. After the adults mated, the females laid eggs on a vertical net, which dropped into distilled water trays. The embryos were collected from the water and placed on an artificial diet for larvae placed in a Petri dish (400 mL distilled water, 10 mL cholesterol, 8.5 mL benzoic acid, 2.5 mL hydrochloric acid, 40 g paper, 30 g sugar, and 30 g yeast powder). The third instar larvae jumped from the Petri dish and pupated in sand, within a box. We transferred the pupae to Petri dishes until their emergence.

### 2.2. RNA Extraction from Adult Flies

Total RNA was extracted from individual flies using TRIzol™ Reagent (Thermo Fisher Scientific, Waltham, MA, USA) according to the manufacturer’s instructions and stored at −80 °C. The concentration and purity of the extracted RNA were determined by measuring the absorbance ratio at 260/280 nm using Thermo Scientific Nanodrop 2000c (Thermo Fisher Scientific, Waltham, MA, USA). Contaminating genomic DNA was removed using an RNase-free DNase I (NEB, Ipswich, MA, USA) treatment and further controlled by RT-PCR using an intron-containing *CcSOD* gene.

### 2.3. Selection of Target Genes, Primer Design, Testing, Sequencing, and Phylogenetic Analyses

A BLASTp search of the NCBI *C. capitata* protein Database, using the protein sequences of *B. tabaci v-ATPase A* ([26]; GenBank: QHB15556.1), *B. tryoni RNase1* and *RNase2* ([33]; XP_039968826.1 and XP_039967124.1), led us to select our study’s orthologous proteins XP_004533376.1, XP_004530585.1, and XP_004530581.1 (named, respectively, as *ATPaseA-CcVha68-1*, *CcdsRNase1* and *CcdsRNase2*), showing, respectively, 90%, 70% and 64% amino acid identity.

Amino acid sequences of *C. capitata* Vha68-1, dsRNase1 and dsRNase2 were used as queries in BLASTp and tBLASTn analyses to identify homologous sequences across various insect orders. The resulting amino acid sequences were aligned using ClustalW, and the alignments for Vha68-1 and dsRNases were used to construct neighbor-joining (NJ) trees with 500 bootstrap replicates with the MEGA X software (version 11.0.13) [45]. Due to the low conservation of dsRNases sequences from positions 1 to 250, the alignment was trimmed to exclude poorly aligned regions.

Subsequently, we designed primers (see Appendix A) to amplify corresponding cDNA fragments (about 500 bp long) by polymerase chain reaction (PCR) derived from the three *C. capitata* orthologous transcripts, also using the *Medfly* genome sequence [44]. We used Primer3 Input software (https://primer3.ut.ee/; accessed on 1 May 2023) with the following parameters: a length of 20 bp, a Tm of 60 °C, and a similar GC content (approximately 45–55%). cDNAs were produced using LunaScript^®^ RT SuperMix Kit (NEB, Ipswich, MA, USA). For sequencing analyses and dsRNA synthesis, the PCR was performed on cDNAs with Phusion^®^ High-Fidelity DNA polymerase (NEB, Ipswich, MA, USA). The obtained amplicons were purified using the Monarch PCR & DNA Cleanup Kit (NEB, Ipswich, MA, USA), according to the manufacturer’s instructions, and sequenced by the Sanger method (Eurofins) (dsRNase1: 613 bp, dsRNase2: 557 bp, and dsATPase: 553 bp). The subsequent alignments using the “MuscleWS alignment” tool (version 5.2) showed 98% identity (7 mismatches, 2 gaps) to *v-ATPaseA-Ccvah68-1* (LOC101448474- XM_004533323.4) compared to the predicted sequence (Appendix A); 99% identity (1 mismatch, 0 gaps) for *CcdsRNase1* (LOC101448568- XM_004530528.2) compared to the predicted sequence (Appendix A); and 99% identity (2 mismatches, 1 gap) for *CcdsRNase2* (LOC101463362- XM_004530524.3) compared to the predicted sequence (Appendix A). The sequencing analyses confirmed that the primers designed for this experiment produce amplicons whose sequences, for each target, correspond to those expected, with identity percentages around 98–99%. The differences could be due to the genetic variability of the strain we use (Benakeion) compared to sequences in the NCBI database derived from the ISPRA strain [44]. After the in vitro synthesis of dsRNA, using the MEGAscript^®^ RNAi Kit (Thermo Fisher Scientific, Waltham, MA, USA), according to the manufacturer’s instructions, for each selected target (*CcVha68-1*, *CcdsRNase1,* and *CcdsRNase2*), the agarose gel showed for each dsRNA a single band of the expected size, suggesting the absence of any off-target effects, and thus indicating their suitability for gene silencing experiments on adult individuals (Appendix A).

### 2.4. Semiquantitative Analysis of Gene Expression

In total, 1 µg of DNase-treated RNA for each sample (whole body, dissected head, thorax, and abdomen) was reverse-transcribed using the LunaScript^®^ RT SuperMix Kit (NEB, Ipswich, MA, USA), according to the manufacturer’s instructions. For semiquantitative analyses, the PCR was performed with One Taq 2X Master mix (NEB, Ipswich, MA, USA) on cDNA, according to the manufacturer’s instructions. cDNAs were used as templates for PCR, using specific pair oligos for each target gene (see Appendix A). After 25 cycles, the individual amplicons were visualized by agarose gel electrophoresis (1.5%).

### 2.5. dsRNA Feeding

Pupae were individually separated into 24-well plates, and adults emerged within 12 h and were transferred into Petri dishes (4 flies for each Petri dish) with perforations for gas exchange. A preliminary test showed that a drop of 10 µL of water–sugar (10%) solution added twice daily ensured 100% adult survival after seven days. Visual inspection showed that all four flies in each Petri dish simultaneously consumed the water drop in 10–15 min. By this approach, we can deduce that, assuming an equivalent feeding ability, on average, each of the four flies ingested 5 µL of solution a day (2.5 µL in the morning and 2.5 µL in the afternoon). One-day and three-day feeding experiments were conducted in parallel. Four feeding/co-feeding groups were set for each experiment: (1) *dsRNases* (dsRNase1, 100 ng/µL; dsRNase2, 100 ng/µL); (2) *dsATPase* (dsATPase, 200 ng/µL); (3) the mix of the two groups *dsRNases/dsATPase* (dsRNase1, 100 ng/µL; dsRNase2, 100 ng/µL; dsATPase, 200 ng/µL); and (4) the control (dsRNA-*GFP*, 200 ng/µL (see Appendix A). In each group, we analyzed 4 flies (4 biological replicates). A total of 16 flies in the first experiment (1 day) and 16 flies in the second parallel experiment (3 days) were sacrificed, respectively, in the afternoon of the second and fourth days to individually extract total RNA and perform qPCR on the four genes, including the housekeeping. The silencing effect was compared between one and three days of feeding.

To evaluate the lethal effect of the *ATPase* mRNA silencing by feeding and co-feeding with dsRNAs, we conducted the experiment as previously described by using the three-day feeding approach (see Appendix A). We fed/co-fed groups of eight newly emerged adult flies (divided into two subgroups of four flies each) with each of the four dsRNA treatments for thirty-two flies. We performed this experiment in three biological replicates, including 96 flies (Appendix A, columns of the three replicate experiments).

After three days of drop feeding, flies were transferred to a small cage and fed with an artificial adult diet and water; the survival rate was monitored over seven days (see Insect Rearing).

### 2.6. Real-Time Quantitative PCR (qPCR)

RNA of each biological replicate (we consider the single fly a biological replicate) was diluted to a 50 ng/µL concentration for qPCR analyses. For each experiment (one-day feeding, three-day feeding, and evaluation of lethality after *ATPase* gene silencing), quantitative real-time analysis (QuantStudio5 Real-Time PCR system, Applied Biosystems, Carlsbad, CA, USA) was conducted to measure the impact of dsRNA treatment on the expression of investigated target genes (*CcdsRNase1*, *CcdsRNase2*, and *CcVha68-1*) by using specific primers listed in Appendix A and considering the housekeeping gene *RpL19* as an internal reference (Appendix A; [46]). Sagri et al. [46] performed extensive qPCR comparative analyses of nine common housekeeping genes (HKGs) of *C. capitata* and the other Tephritidae *B. oleae* in various tissues and developmental stages. They found the *RpL19* gene to be an appropriate HKG for standardization and control reference, which we used in our study.

The relative expression of genes was measured by one-step qPCR, using the Power SYBR™ Green RNA-to-CT™ 1-Step kit (Applied Biosystems, Carlsbad, CA, USA), according to the manufacturer’s instructions. Gene expression data were analyzed using the 2^−ΔΔCT^ method [47,48]. For method validation, the difference between the Ct value of one of the selected target genes and the Ct value of the reference gene *Rpl19* [∆Ct = Ct (target) − Ct (reference)] was measured against serial dilutions (400 ng, 200 ng, 100 ng, 50 ng, and 25 ng) of the extracted RNA. The equation of the obtained curve (for each target/reference pair) had a slope value of less than 0.1, indicating that primer efficiencies were approximately equal.

### 2.7. Ex Vivo Nuclease Activity Degradation Assay

To assess the impact of gut nucleases on dsRNA molecules, we followed a previous protocol [33].

Eight flies (four males and four females) were fed with dsRNA drops (eight flies fed *dsGFP* and eight flies fed *dsRNases*) for three days, as previously described. On the fourth day, the flies were dissected in phosphate-buffered saline (PBS 1X) to collect the gut from the mouth to anus, excluding the crop and Malpighian tubules. The single dissected eight guts were placed in Eppendorf tubes and resuspended in 10 μL of PBS 1X. After a mild centrifugation step at 3000× *g* for 5 min to allow for the release of the gut juice, the supernatants were collected and refrigerated at 4 °C for 16 h to enable enzymes to diffuse into the solution. Then, 100 ng of *dsATPase* was added to a diluted solution (1:20) of supernatant and incubated at RT for four time points (0, 15, 30, and 60 min). Samples from each time point were stored at −20 °C until visualization using 1.5% agarose gel electrophoresis. Band fluorescence intensities were measured using Image Lab™ software (version 6.1.0.7) (Bio-Rad, Hercules, CA, USA).

### 2.8. Statistical Analysis

The normality of the data obtained through real-time analysis was assessed using the Shapiro–Wilk and Kolmogorov–Smirnov tests. The expression levels of the target genes in different biological groups were analyzed using the one-way ANOVA test. When significant effects were observed (*p*-value < 0.05), Dunnett’s test was used for multiple comparisons. Survival curves of the different biological groups analyzed after dsRNA treatment were compared using the log-rank test (Mantel–Cox). The ex vivo degradation assay data were analyzed using the two-way ANOVA test, as affected by dsRNA treatment and time post-treatment. All statistical analyses were performed using GraphPad Prism 9.0 (GraphPad Software, San Diego, CA, USA).

## 3. Results

### 3.1. Homology Search Led to the Discovery of Three ATPase and Two dsRNase Orthologues in the Medfly Genome

We found three *C. capitata* orthologous v-ATPase A proteins by BLASTp analysis and selected the first hit for our study. Based on the *Drosophila* Flybase nomenclature of three related v-ATPase A proteins, we named this first gene *CcVha68-1* [49]. 

The *B. tryoni* dsRNase1 (XP_039968826.1; LOC120780634; BtdsRNase1) and dsRNase2 orthologous proteins (XP_039967124.1; LOC120779055; BtdsRNase2) used in the Tayler et al. [33] study show 49% aa sequence identity. A BLASTp search in *C. capitata* found two paralogous proteins also showing 48% aa identity (XP_004530585.1, XP_004530581.1). We concluded that they are the corresponding orthologues of the *B. tryoni* dsRNase1 and dsRNase2.

The neighbor-joining (NJ) tree for *Vha68-1* (Figure 1) shows a clear correspondence between taxonomic orders and clades, with most groups being statistically well supported (e.g., Diptera, Lepidoptera, and Coleoptera, each with 100% bootstrap support).

The NJ tree for *dsRNases* (Figure 2) reveals the presence of two paralogs (*dsRNase1* and *dsRNase2*) only in Diptera, while species from other examined orders possess a single *dsRNase*. This gene duplication appears to be a unique feature of Diptera, although further and more extensive analyses are required to confirm this finding. Like the *Vha68-1* tree, the *dsRNases* tree also shows groupings that correspond to taxonomic orders. However, some discrepancies are observed in Orthoptera and Hemiptera compared to the *Vha68-1* tree, likely due to the limited number of sequences from these orders in our analyses.

We designed and transcribed in vitro 0.5 kb long dsRNAs targeting the three gene functions. The three DNA templates for in vitro transcription were obtained by RT-PCR, gel-purified and sequenced. The *CcVha68-1* dsRNA sequence shows an overall 78% nucleotide identity with the paralogue *CcVha68-2*, with only two short regions of 21 bp and 68 bp having 100% identity. The two *CcdsRNases* sequences show 69–72% DNA identity over 103–137 long nucleotide (nt) regions of the corresponding paralogues, with the most extended stretches being 14 nt. This analysis suggests that the three dsRNAs (*dsATPase, dsRNase1*, and *dsRNase2*) are not expected to cause significant intergenic RNAi on the respective *C. capitata* paralogues.

### 3.2. Semiquantitative Analysis Showed dsRNases Highly Expressed in the Abdomen

We assessed the gene expression of the three *C. capitata* orthologues in dissected heads, thoraxes, and abdomens of adult flies by a semiquantitative RT-PCR (Figure 3). We found that the *CcVha68-1* gene is expressed in all three body parts (Figure 3a), while we observed high expression of both dsRNases in the abdomen, and low (*CcdsRNase1*) or no expression (*CcdsRNase2*) in the head or thorax, suggesting conservation of their gut-specific (*CcdsRNase2*) or gut-biased (*CcdsRNase1*) expression (Figure 3b,c).

### 3.3. Gene Silencing Analysis Revealed Enhanced RNAi Effect after Three Days of Co-Feeding

We investigated the effects of dsRNA feeding on the three targeted genes by qPCR after one day and three days of treatments. After one and three days of feeding treatments, a reduction of 50% was observed in *CcVha68-1* mRNA levels in flies fed with dsATPase alone (Figure 4a,b) (one-way ANOVA: *p* < 0.001). In the two one-day and three-day feeding experiments, respectively, a 60% and 70% reduction in both *CcdsRNase1* and *CcdsRNase2* mRNA levels were observed in dsRNases and mix groups (Figure 4c–f) (one-way ANOVA: *p* < 0.0001). Interestingly, 50% and 75% *CcVha68-1* mRNA reductions were observed after one day and after three days in flies fed with the mix (*dsRNases* + *dsATPase*). Hence, after three days of co-feeding, the gene silencing of *CcVha68-1* was improved by 20–25% in the mix group compared to dsATPase alone (Figure 4b) (one-way ANOVA: *p* < 0.0001). This effect, however, is not observable when the flies are fed and analyzed after one day (Figure 4a) (one-way ANOVA: *p* < 0.001). This suggests that in *C. capitata*, as also observed in *B. tryoni* [33], simultaneous silencing of the two intestinal nucleases (*CcdsRNase1* and *CcdsRNase2*) favors an increase in the molecular efficiency of RNAi against a third target gene, at least after three days.

### 3.4. Co-Feeding dsRNA Targeting Gut Nucleases and a Vital Gene Led to a More Effective Mortality

Based on these silencing expression data, we explored the potential mortality effects after a three-day feeding experiment over seven days.

We compared mortality rates at days five and seven following the first three days of drop feeding with the various dsRNA solutions (Figure 5). *dsGFP* had no substantial lethal effect on day five (1/24 died; 4%) or day seven (2/24 died; 8%) (Figure 5, green line). Similarly, dsRNases feeding led to 8% (2/24) and 16% (4/24) mortality rates (Figure 5, blue line). dsATPase induced a higher mortality of 29% (7/24) and 37% (9/24), respectively (Figure 5, red line). The mix (co-feeding with *dsRNases * + *dsATPAse*) was much more effective, inducing a mortality of 46% (11/24) and 79% (19/24), respectively (Figure 5, violet line). If we add the mortality percentages at day seven of the two single treatments with dsRNAses (17%) and dsATPase (37%), we obtain a 53% mortality. The observed 79% mortality with the dsRNA mix suggests its synergistic rather than additive lethal effect. Comparing the triplicates, the mix induced the death of 8/8, 6/8, and 5/8 flies, respectively (the first two experiments were carried out with the same dsRNA batch). Based on our and previous data [33], we speculated that the silencing of gut nucleases prevented, at least partially, the degradation of the dsRNA *ATPase* (*CcVha68-1*) molecules and boosted its silencing action on the endogenous vital function.

To further control the molecular efficacy of *CcVha68-1* silencing in the four dsRNA feeding groups, we repeated, in parallel to the mortality experiments, the qPCR on an additional 16 flies (sacrificed at day four) in biological duplicates (a total of 32 flies). As in the previous qPCR, we confirmed a *CcVha68-1* mRNA reduction of 75% (Appendix A, 4th column) (one-way ANOVA: *p* < 0.0001).

### 3.5. Nuclease Gene Silencing Partially Prevents Degradation of dsRNA within the Gut Juice

To test how silencing intestinal nucleases protects dsRNA from degradation, we performed an ex vivo assay. In all replicates, the dsATPase was degraded entirely (100%) within 60 min when incubated in the gut juice of dissected *dsGFP*-fed adults, indicating no protection from degradation. In contrast, in the dsRNases group, dsATPase was partially degraded (about 70% of degradation) within the same time interval (two-way ANOVA: *p* < 0.0001), indicating a 30% protection efficacy (Figure 6b). At 30 min, in the dsRNases-fed adult guts, the protection efficacy of dsATPase is 70%, while in the *dsGFP*-fed, protection efficacy is only 20% (Figure 6b). These data suggest that the gene silencing of both gut nucleases by co-feeding significantly slows down the *CcVha68-1* dsRNA degradation in vivo, improving both the *CcVha68-1* mRNA reduction and the related lethal effect.

## 4. Discussion

Climate change poses new risks of alien pest insect invasion and expansion into new territories [50]. Furthermore, insecticide resistance is becoming a serious problem also in *C. capitata* [51,52]. The European Green Deal has as a milestone the so-called Farm to Fork Strategy, aiming to develop novel food systems that are fair, healthy, and environmentally friendly (https://food.ec.europa.eu/horizontal-topics/farm-fork-strategy_en; accessed on 1 July 2024). This strategy sets out a key objective: a 50% reduction by 2030 in the use and risk of chemical pesticides. The development of novel solutions, including biopesticides based on dsRNAs, is urgent but limited in many cases. It is also due to the identification of very effective vital gene targets and the degradation of dsRNAs in the insect gut. The limitation of the dsRNA production costs seems to have been overcome recently, as commercial dsRNA products such as Ledprona (dsRNA-specific for a proteasome subunit of *Colorado potato beetle L. decemlineata*; EPA-registered, Washington, DC, USA) and the Ledprona-based Calantha spray formulation (GreenLight Biosciences, Durham, NC; EPA-submitted, Washington, DC, USA) [53] are close to reaching the market.

Searching for vital gene targets expressed in insect guts in the last few years led to identifying some candidates. dsRNA feeding to target these genes led to a high degree of mortality only in a few cases, including, for example, one hemipteran (*B. tabaci*) and three coleopteran species (*Diabrotica virgifera virgifera*, *L. decemlineata*, and *Anthonomus grandis*) (Table 1). Upadhyay et al. [26] caused 97% mortality in *B. tabaci* (Hemiptera) adult white flies (piercing–sucking mouth parts adapted for feeding on plant sap) after six days of ad libitum dsRNA *ATPase A* (*Drosophila Vha68-1* orthologue; see results) feeding with liquid diet (20 ng/µL; adult size is small, only 1–2 mm) (Table 1). Rangasamy and Siegfried [22] fed the coleopteran *D. virgifera virgifera* (chewing mouthparts adapted for biting and grinding solid plant tissues) with an artificial semi-solid *ATPase* dsRNA diet (diet plug 4 mm; 2 mm for each adult, containing 1000 ng dsRNA every three days), causing 95% mortality after two weeks (Table 1). On the contrary, in the Tephritidae *B. dorsalis*, 14 days of long continuous feeding of dsRNA or dsRNA-expressing bacteria to target *v-ATPase D* induced only a 30% mRNA decrease but no mortality [28] (Table 1). In the mosquitoes, *Aedes aegypti*, oral delivery of *v-ATPase A* dsRNA (10 mosquitoes fed with 1000 ng/µL 10% sucrose ad libitum for 24 h) failed to induce mortality after 48 h (Table 1). Ortolà et al. [30] used novel circular dsRNAs to target *v-ATPase* (*Vha68-2* orthologue). Injection of circular dsRNA (500 ng) into adult flies resulted in approximately 70% mortality within one week, with a molecular gene silencing of about 50–60%. Conversely, oral administration (20 adults in triplicates fed with a single 10 µL droplet of 1 µg/µL of dsRNA solution of 30% sucrose for three days; theoretically, each fly ingested a total of 1.5 µg in three days) led to non-significant mortality (15%) compared to the control, but to a 48% reduction in mRNA levels (Table 1). In summary, the mortality rates induced by feeding with dsRNA targeting *v-ATPase* subunits can vary dramatically (0–97%); generally, the rates are higher in Hemiptera and Coleoptera than in Diptera.

Our study was inspired in part by Upadhyay et al. [26], which identified and targeted by oral RNAi the white fly *B. tabaci* (Hemiptera) *v-ATPase A* gene (*BtATPase*). Hence, we selected the *v-ATPase Drosophila Vha68-1* orthologue in *Ceratitis capitata* as a target for our experiments. Similarly to *D. melanogaster* (based on expression data available at Flybase, https://flybase.org/; accessed on 1 July 2024), the *CcVha68-1* gene is expressed in many adult body tissues of *Medfly*, including the head, the thorax, and the abdomen, suggesting a likely ubiquitous expression.

When we fed *C. capitata* adults for three days with dsRNAs targeting only *CcVha68-1* (expected on average 3 µg ingested dsRNA for each fly), we observed over one week a 37% mortality rate, in contrast to the non-significant mortality observed by Ortolà et al. [30] (Table 1). This difference could be due to different reasons: (1) the targeting of different *v-ATPase A* genes, (2) the use of circular versus linear dsRNAs, (3) the amount of delivered dsRNA, (4) differences in the strains of *C. capitata* (it would be a useful strategy also to exchange *C. capitata* strains among the two laboratories), and (5) the oral administration protocol: feeding four flies twice a day for three days with a 10 µL droplet (200 ng/µL dsRNA). Ortolà et al. [30] used 20 flies and a 10 µL droplet once a day (1 µg/µL dsRNA) for three days. It would be interesting to test, for example, circular dsRNA targeting *CcVha68-1* instead of *Vha68-2* to investigate if the lack of mortality was due to a different target. For example, Taning et al. [11] found that feeding *D. suzukii* adult flies with dsRNAs targeting the *Vha26* subunit induced low mortality values (Table 1).

Interestingly, Whyard et al. [27] targeted, by dsRNA, oral feeding (up to 3 µg/µL, 20% sucrose ad libitum for one week) of the E-subunit of the *v-ATPase* gene in a broader range of insects, including larvae of a beetle (*T. castaneum*), moth larvae (*M. sexta*), aphid nymphs (*A. pisum*), and dipteran larvae (*D. melanogaster*; these larvae were soaked with liposome dsRNA for only 2 h). These authors observed a 50–75% mortality rate over one week (Table 1). It will be interesting to investigate if *v-ATPase E*-subunit dsRNA feeding has a similar high efficacy in adults of *D. melanogaster* and of other dipteran species, including *C. capitata*, as well as to combine the dsRNA protection also with nuclease silencing.

Protection of orally supplied dsRNA can also be achieved by suppressing the degrading activities of intestinal nucleases, which are among the major causes of differential RNAi efficiency reported among insects [40]. Tayler et al. [33] delivered dsRNAs (1 µg) targeting *dsRNase-1* and *-2* into adults of the other Tephritidae *B. tryoni* using injections because RNAi is more efficient than feeding; they also performed an ex vivo assay (Table 1). The gut juices from the untreated flies and injected flies digested 100% and 30% of a target dsRNA (750 ng) in vitro in 1h, revealing a 70% increase in the protection from dsRNA degradation. We preferred feeding adult flies with dsRNAs rather than injecting them and monitoring *CcVha68-1* dsRNA degradation to simulate more realistic conditions.

Within the *C. capitata* genome, we found two nucleases orthologous to those described in *B. tryoni* and other species [33]. As in *B. tryoni*, *CcdsRNase1* and *CcdsRNase2* seem to be expressed either predominantly (*CcdsRNase1*) or exclusively (*CcdsRNase2*) in the abdominal region (likely the guts).

Our ex vivo assay showed that the *C. capitata* gut juices from untreated flies and dsRNases-fed flies digested 100% and 70% of 100 ng of dsRNA in 1 h (in a test tube). These data showed that the co-silencing of two nucleases by dsRNA feeding increased by 30%, protecting dsRNA from degradation. The lower efficiency in protection with respect of the Tayler et al. [33] study is likely due to increased RNAi efficiency after the injection rather than feeding.

A second analysis that can be performed to investigate the effect of the nuclease silencing is to measure by qPCR the mRNA levels of a third target gene. Tayler et al. [33] co-fed for six days with a dsRNA mix (2 dsRNases + dsRNA-*yellow*) a group of ten flies and induced a 100% reduction in the *yellow* RNA levels compared to *GFP* dsRNA feeding. Oral feeding with only dsRNA-*yellow* induced a 70% reduction (see Figure 3c in [33]). Our data showed that feeding with only dsATPase induced a 50% reduction in *CcVha68-1* mRNA levels, while co-feeding with dsRNA mix (2 *dsRNases + dsATPase*) induced a 75% reduction in *CcVha68-1* mRNA levels (compared to *GFP* dsRNA feeding). In both studies, we can conclude that the co-silencing of intestinal nucleases improved the molecular silencing of a third gene by 25–30%.

The final experiment of our study was to measure the mortality rate induced by co-feeding with dsRNA silencing of the two *dsRNases* and the *ATPase A* subunit (*CcVha68-1*).

We decided to perform a three-day co-feeding experiment with only two droplets of 10 µL every day (200 ng/µL) to maintain the potential lower future costs of a biopesticide application, instead of ad libitum feeding during a 1–2 week period, as used in other studies, targeting only the *v-ATPase* gene function [22,26,27,28]. Similarly, Ortolà et al. [30] fed *C. capitata* flies once a day for three days, monitoring the mortality over one week. 

Our data showed that co-silencing the three genes by adult feeding, compared to silencing only *CcVha68-1*, improved the mortality rate by 42% (from 37% to 79%) in a week. This observation is coherent with the previous ex vivo experiment in which nuclease silencing protected the target dsRNA from degradation (an increase of 30% after 1 h), and with a 75% reduction in the *CcVha68-1* mRNA.

Our data confirm that intestinal nuclease activity reduces the efficacy of oral RNAi [31,33,36,40,54,55,56,57]. A simple addition of the mortality percentages on day seven of the two single treatments with dsRNases (17%) and dsATPase (37%) leads to a 53% mortality rate. The 79% mortality rate we observed using the dsRNA mix suggests a synergistic rather than additive lethal effect of co-feeding, and that dsRNases co-silencing improved the mortality rate by 25%.

Also, Spit et al. [32] silenced two intestinal dsRNases and a vital gene (the authors failed to provide the DNA sequence of this gene, named as *Ld_lethtgt*) by oral feeding simultaneously in the coleopteran *L. decemlineata* (potato pest; Colorado Potato Beetle, CPB; chewing mouthparts to feed on solid plant tissues) (Table 1). They fed pre-pupal CPB with one dose of 1400 ng of dsRNases1/2 and repeated the single-dose feeding post-emergence in the adults, adding 500 ng dsRNA of *Ld_lethtgt* vital gene. They observed that after eight days, there was 82% mortality. The adults fed with a single dose of dsRNA (500 ng) targeting only the vital gene led to 75% mortality. Hence, compared to our data, they observed only a mild improvement in mortality (7%) when co-silencing the intestinal nucleases.

Similarly, Almeida Garcia et al. [31] micro-injected dsRNAs targeting three dsRNA nucleases (500 ng for each in a sucrose 5% water drop) into young adults of the cotton boll weevil (*A. grandis*; Coleoptera; piercing–sucking mouth); after two days of starvation, they fed them with dsRNA targeting the vital *Chitin synthase II* (500 ng in a sucrose 5% water drop) (unfortunately, these authors missed performing a final co-feeding experiment instead of microinjections to render their method more applicable in the field) (Table 1). They observed after ten days that the nuclease suppression (injection at adult stages) led to 15% mortality, while *Chitin synthase II* dsRNA feeding led to 60% mortality; the combined mortality of single dsRNA treatments was 75%. The application of both silencing practices led to an increase of only 10%, reaching 85% in mortality, while we observed a 25% increase in our experiments.

The cause of death in insects following the oral delivery of dsRNA targeting *ATPase* could be due to the critical reduction in ATP production, leading to cellular energy failure. The latency in the onset of mortality (few days) can be explained, for example, by the time required for the RNAi process to effectively reduce *ATPase* levels, by the gradual depletion of ATP, by the individual variability in gene knockdown efficiency/speed, and by possible compensatory mechanisms within the insect’s cells.

## 5. Conclusions

We conclude that our feeding RNAi strategy achieved a higher and faster mortality rate for the first time in a Tephritidae species, close to those achieved only in coleopteran and hemipteran species [9,26] (Table 1). We will explore how to achieve a higher mortality rate shortly. We will modify some of the parameters of the applied method: (1) extend the feeding time; (2) increase dsRNA quantity and concentration; (3) use nanoparticles and liposomes, either as substitutes of dsRNases silencing or in conjunction with them; and (4) select novel intestinal vital target genes. We have already identified potential new gene targets based on several works from the literature: *Putative COPI coatomer* gene, *Chitin synthase II* gene, and other subunits of the *v-ATPase* gene.

It is desirable to develop a standard protocol of oral RNAi to be applied as a control reference and to more reliably compare different studies in the insect RNAi community. For example, in our protocol, we can approximately measure the quantity ingested by each fly and the time required for ingestion. In parallel, a second protocol could be designed that mimics more real field applications once more efficient dsRNA combinations and delivery methods are found. Furthermore, improved nomenclature and gene models of the promising *ATPase A* subunits and *dsRNases* in the various Tephritidae species, including paralogues and phylogenetic comparisons, are required to better define the chosen paralogue targets in future studies (Volpe et al., manuscripts in prep).

## Figures and Tables

**Figure 1 insects-15-00717-f001:**
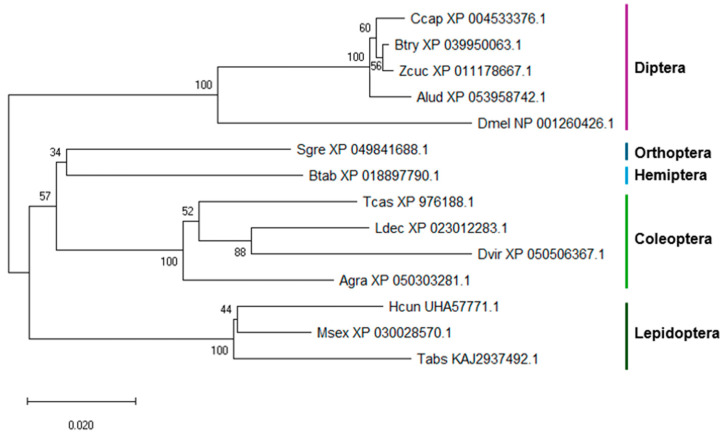
NJ tree of the Vha68-1 proteins. The accession numbers of the sequences are shown near the abbreviation. The bootstrap support percentages are shown next to the branches. Ccap, *Ceratitis capitata*; Btry, *Bactrocera tryoni*; Zcu, *Zeugodacus cucurbitae*; Alud, *Anastrepha ludens*; Dmel, *Drosophila melanogaster*; Sgre, *Schistocerca gregaria*; Btab, *Bemisia tabaci*; Tcas, *Tribolium castaneum*; Ldec, *Leptinotarsa decemlineata*; Dvir, *Diabrotica virgifera virgifera*; Agra, *Anthonomus grandis*; Hcun, *Hyphantria cunea*; Msex, *Manduca sexta*; Tabs, *Tuta absoluta*.

**Figure 2 insects-15-00717-f002:**
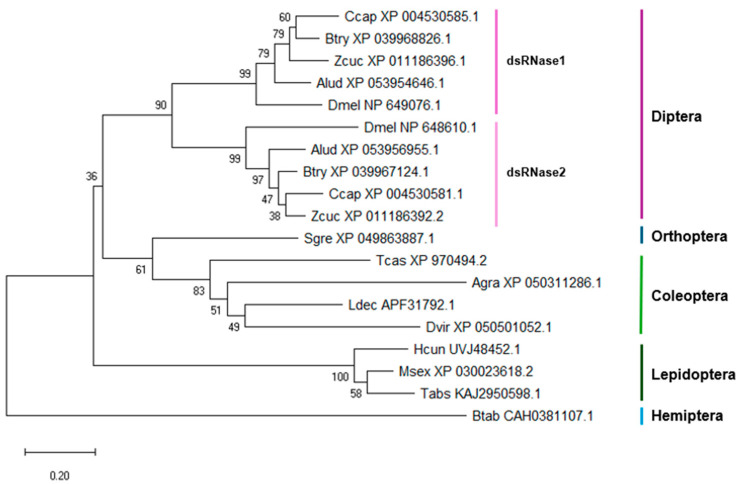
NJ tree of the dsRNases proteins. The accession numbers of the sequences are shown near the abbreviation. The bootstrap support percentages are shown next to the branches. Ccap, *Ceratitis capitata*; Btry, *Bactrocera tryoni*; Zcu, *Zeugodacus cucurbitae*; Alud, *Anastrepha ludens*; Dmel, *Drosophila melanogaster*; Sgre, *Schistocerca gregaria*; Btab, *Bemisia tabaci*; Tcas, *Tribolium castaneum*; Ldec, *Leptinotarsa decemlineata*; Dvir, *Diabrotica virgifera virgifera*; Agra, *Anthonomus grandis*; Hcun, *Hyphantria cunea*; Msex, *Manduca sexta*; Tabs, *Tuta absoluta*.

**Figure 3 insects-15-00717-f003:**
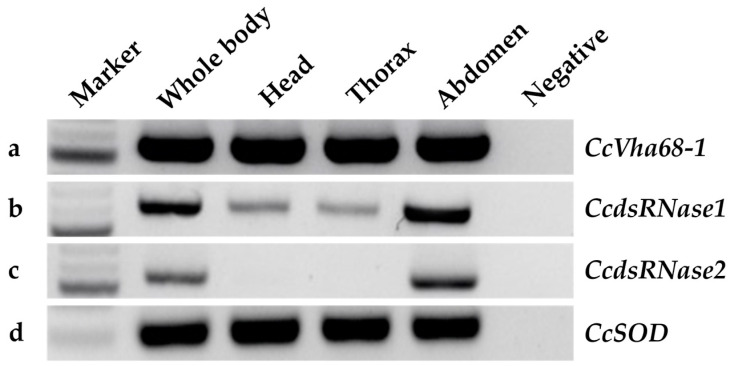
Semiquantitative analysis of *CcVha68-1*, *CcdsRNase1*, *CcdsRNase2* and *CcSOD* genes on cDNA from whole body, head, thorax, and abdomen. (**a**) *CcVha68-1* expression shows a 553 bp band in all samples; (**b**) *CcdsRNase1* expression shows a 613 bp band in all samples, with higher intensity in the abdomen; (**c**) *CcdsRNase2* expression shows a 557 bp band exclusively in the abdomen; (**d**) *CcSOD* expression (housekeeping) shows a 300 bp band in all samples (the lack of a 128bp-long intron in the amplified *CcSOD* DNA product indicted a gDNA-free cDNA sample; see Appendix A). For original agarose gels, see Appendix A.

**Figure 4 insects-15-00717-f004:**
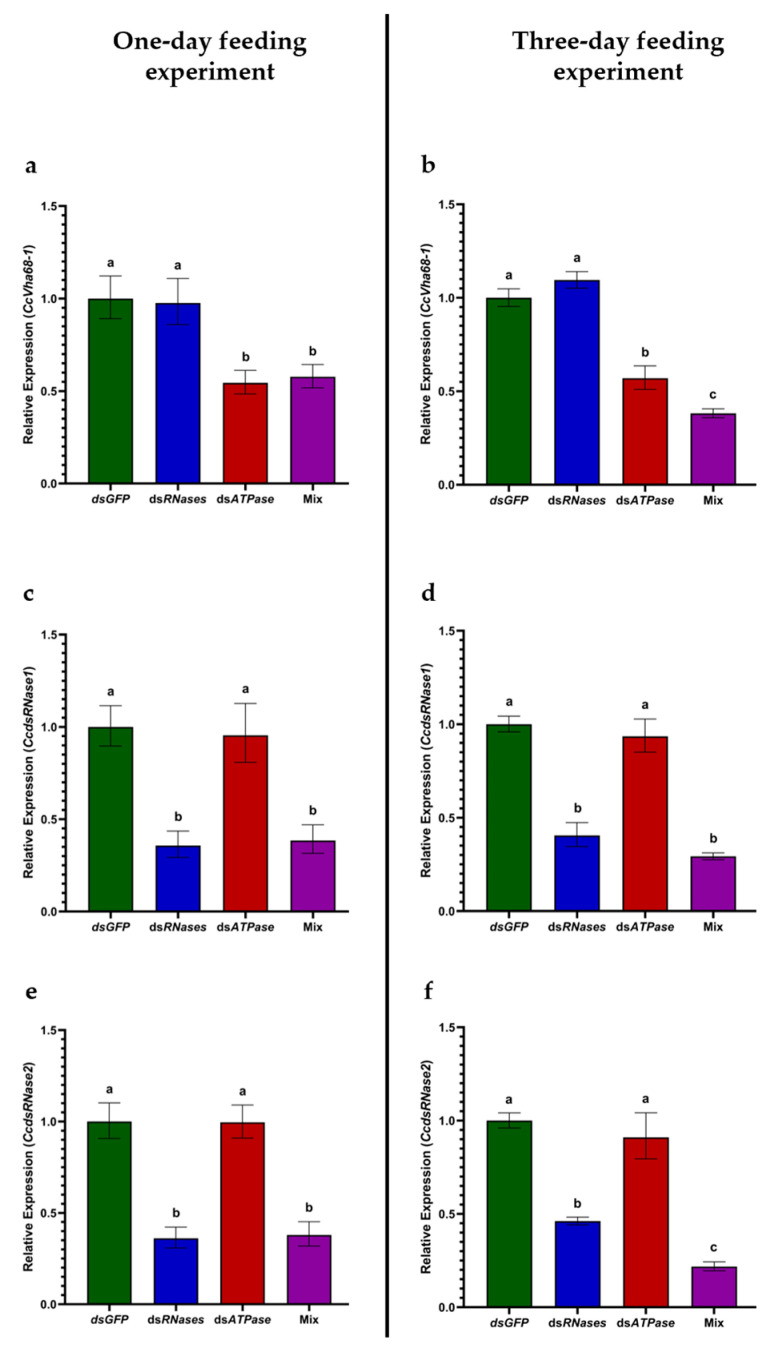
Transcript levels of *CcVha68-1*, *CcdsRNase1* and *CcdsRNase2* post one-day and three-day treatments. Different letters denote a significant difference between mean values recorded for each group (the obtained values passed normality tests). The values reported are the mean ± standard error. (**a**) Relative expression of *CcVha68-1* post one-day treatment (one-way ANOVA: F(3,28) = 7.855, *p* < 0.001); (**b**) relative expression of *CcVha68-1* post three-day treatment (one-way ANOVA: F(3,27) = 47.09, *p* < 0.0001); (**c**) relative expression of *CcdsRNase1* post one-day treatment (one-way ANOVA: F(3,28) = 15.92, *p* < 0.0001); (**d**) relative expression of *CcdsRNase1* post three-day treatment (one-way ANOVA: F(3,28) = 37.9, *p* < 0.0001); (**e**) relative expression of *CcdsRNase2* post one-day treatment (one-way ANOVA: F(3,28) = 24.08, *p* < 0.0001); (**f**) relative expression of *CcdsRNase2* post three-day treatment (one-way ANOVA: F(3,28) = 59.01, *p* < 0.0001).

**Figure 5 insects-15-00717-f005:**
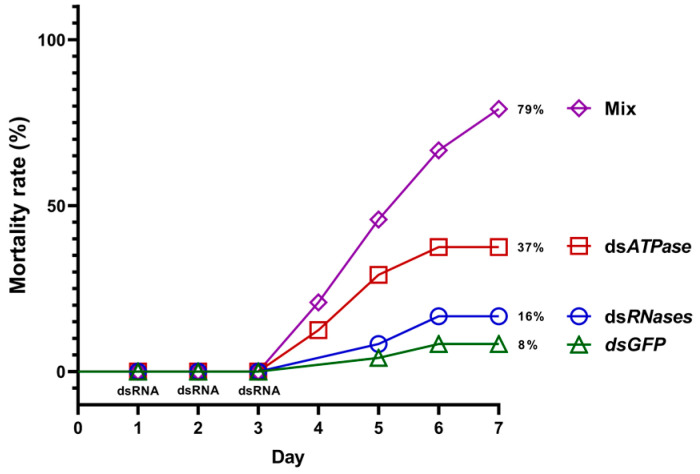
The mortality rate of treated and control flies. (Log-rank (Mantel–Cox) test: χ^2^ = 34.53; df = 3; *p* < 0.0001). Feeding with dsRNAs was performed during the first three days. The mortality percentages are referred to day seven only. We used 24 biological replicates (flies) for each experimental group assessing mortality over seven days.

**Figure 6 insects-15-00717-f006:**
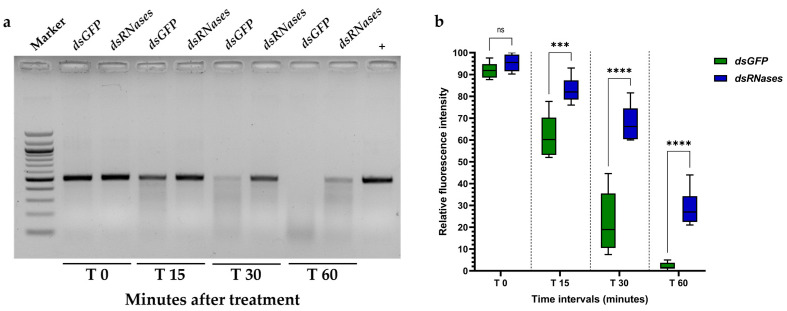
Ex vivo degradation assay of dsATPase molecules (100 ng for each sample). (**a**) An example of one gel electrophoresis analysis of dsRNA degradation at different time points (0, 15, 30, and 60 min) after dsRNA (*dsGFP* and dsRNases groups) treatment (see Appendix A for all agarose gels). The last lane represents positive control of dsRNA with no gut juice. (**b**) Overall trend (*n* = 8 for each group) of nuclease degradation activity at different time intervals after treatment. Asterisks denote a significant difference between mean values recorded for each group (the obtained values passed normality tests): *** = 0.0002; **** = < 0.0001; ns = not significant. The values reported are the mean ± standard error. Both the dsRNA treatment (two-way ANOVA: F(1,14) = 74.34, *p* < 0.0001) and the time post-treatment (two-way ANOVA: F(2.080,29.12) = 554.2, *p* < 0.0001) significantly affected the degradation of dsATPase.

**Table 1 insects-15-00717-t001:** Experiments from the literature on *v-ATPase* and *dsRNases* genes.

Reference	Order—Species	Stage	Targeted ATPase Genes	*n*. of Individuals	Quantity of dsRNA	Mortality %
Whyard et al. [27] (Ins bioch and mol bio)	(Coleoptera)—*T. castaneum* (Lepidoptera)—*M. sexta* (Hemiptera)—*A. pisum* (Diptera)—*D. melanogaster*	Larvae and nymphs	v-ATPase E	N.C.	3 µg/µL ad libitum	50–75% (one week)
Li et al. [28] (Plos one)	(Diptera)—*B. dorsalis*	Adults	v-ATPase D	60 (triplicate)	2 µg/µL	No lethality
Upadhyay et al. [26] (J. Biosci)	(Hemiptera)—*B. tabaci*	Adults	v-ATPase A Drosophila vha68-1 orthologue	20 (triplicate)	20 ng/µL ad libitum	97.5% (after 6 days)
Coy et al. [29] (J. Appl. Entomol)	(Diptera)—*A. aegypti*	Adults	v-ATPase A Drosophila vha68-2 orthologue	10	1000 ng/µL ad libitum	No lethality
Rangasamy and Siegfried [22] (Pest Manag Sci)	(Coleoptera)—*D. virgifera virgifera*	Adults	v-ATPase A Drosophila vha68-2 orthologue	16 (5 replicates)	1000 ng dsRNA every three days	95% (within 2 weeks)
Taning et al. [11] (Journal of Pest Science)	(Diptera)—*D. suzukii*	Adults	v-ATPase E Drosophila vha26 orthologue	32-40 (triplicate)	50 mg of diet mixed with 32 µg dsRNA	10–23%
Ortolà et al. [30] (Pest Manag Sci)	(Diptera)—*C. capitata*	Adults	v-ATPase A Drosophila vha68-2 orthologue	20 (triplicate)	1000 ng/µL for three days	No lethality (reduction in fecundity)
**Reference**	**Order—Species**	**Stage**	**Targeted Gut Nuclease Genes**	** *n* ** **. of Individuals**	**Quantity of dsRNA**	**Effects on Target Gene after Co-Feeding Experiment**
Almeida Garcia et al. [31] (Plos one)	(Coleoptera)—*A. grandis*	Adults	AgraNuc1, AgraNuc2 and AgraNuc3	N.C.	500 ng	Increasing in mortality of about 10%
Spit et al. [32] (Ins bioch and mol bio)	(Coleoptera)—*L. decemlineata*	Larvae and adults	Ld_dsRNase1 and Ld_dsRNase2	N.C.	1400 ng	Increasing in mortality of about 7%
Tayler et al. [33] (Open biology)	(Diptera)—*B. tryoni*	Adults	dsRNase1 and dsRNase2	10	1 μg/μL	Increasing in gene silencing of 30%
Ahmad et al. [34] (Journal of Pest Science)	(Diptera)—*Z. cucurbitae*	Larvae	ZcdsRNase1, ZcdsRNase2	60 (triplicate)	1000 ng/μL	Increasing in gene silencing of about 35–40% and mortality of about 25%

## Data Availability

The authors confirm that the data supporting the findings of this study are available within the article and its Appendix A.

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
