# Peer review of "Simultaneous Silencing of Gut Nucleases and a Vital Target Gene by Adult dsRNA Feeding Enhances RNAi Efficiency and Mortality in *Ceratitis capitata"

_insects, 2024, doi:10.3390/insects15090717_

Round 1
Reviewer 1 Report
Comments and Suggestions for Authors
This manuscript entitled “Simultaneous silencing of gut nucleases and a vital target gene by adult dsRNA feeding enhances RNAi efficiency and mortality in Ceratitis capitata adults” presents an interesting and important investigation of the effects of simultaneous feeding of dsRNAs of gut nucleases and V-ATPase A subunit on enhancing the RNAi effectiveness and mortalities. Although the experimental design is simple, but it provides an exciting discovery about a potential and novel strategy for the sustainable control of this destructive pest, Mediterranean fruit fly. The manuscript content is logic, the results are clearly described, discussion is sufficient. However, the deficiencies in abstract, result, and discussion description were also found as described as followed:
1. The critical discovery of this manuscript is that simultaneous feeding dsRNA of gut nucleases could enhance the effectiveness of RNAi of one vital target gene (v-ATPase A). Based on this discovery, the more important thing is that it shed light in the potential universal effects of feeding dsRNA of gut nucleases on the effectiveness of RNAi of vital target gene(s) for the sustainable control of this destructive pest, Mediterranean fruit fly. However, only one vital target gene (v-ATPase A) is not enough to show the potential universal effects. So I suggest adding more vital target gene(s), or, at least, discussing it as one of important future works.
2. Title: delete the last “adults”
3. Abstract: 1) the author said that “…which is poorly explored in dipteran species” in line 34. As I known, feeding dsRNA for RNAi have been explored in some labs. However, there was very few reports mentioning it because of its low effectiveness. Therefore I suggest to replace it by "…which is poorly reported in dipteran species". 2) in lines 40-41, the author said that “In contrast, we observed 79% mortality over seven days, which was associated with a decrease in mRNA levels of the three targeted genes”; I wonder what the treatment is; feeding dsRNA mixture of three genes?
4. Introduction: 1) the introduction could be more concise and fucus on “simultaneous RNAi”, “v-ATPase”, and “gut nucleases”. For example, in the second paragraph, the author listed a lot of RNAi effects on various target genes, which were not necessary. While in the six paragraph, few information of “gut nucleases” was showed. 2) in line 80, the author mentioned the “systemic RNAi” in B. dorsalis; however, I could not find any evidence (even from the ref 14 cited by author) to support it.
5. Results: 1) as usual, the subsection titles of the “Results” should show the major related result of each part to facilitate reading, rather than introduce the research contents, please improve it. 2) the detailed method description and related information should not be shown in “Results”; this can facilitate the reader focusing on the major results; for example, in lines 322-332, the author introduced the detailed information of housekeeping gene selection and dsRNAs feeding experiments, which is better to be combined into “Materials and Methods”
6. Result “3.1”: As usual, the phylogenetic analysis was used to show the orthologous relationship of specific gene(s) to confirm the annotation of the gene sequence(s). I suggest the author add it.
7. Discussion: this part is lengthy and not very readable; for example, paragraphs 7-12 (lines 519-566) just concluded the results of RNAi effects on mRNA levels and mortalities and compared to previous related reports, which is better to be combined into one paragraph with more concise description.
8. Reference: The format of the text must be unified. In addition, the species scientific names should be italicized (such as in ref 13)
9. Supplementary materials: 1) the supplementary tables could not be found; 2) figure S1 – S3 could not be found, and the basic information of each lane in each figure of electrophoresis gel should be added.
Comments on the Quality of English LanguageQuality of English Language is good
Author Response
- The critical discovery of this manuscript is that simultaneous feeding dsRNA of gut nucleases could enhance the effectiveness of RNAi of one vital target gene (v-ATPase A). Based on this discovery, the more important thing is that it shed light in the potential universal effects of feeding dsRNA of gut nucleases on the effectiveness of RNAi of vital target gene(s) for the sustainable control of this destructive pest, Mediterranean fruit fly. However, only one vital target gene (v-ATPase A) is not enough to show the potential universal effects. So I suggest adding more vital target gene(s), or, at least, discussing it as one of important future works.
Thanks for the comment. Although our intentions were not to demonstrate this (improvable) method as universally usable in enhancing the RNAi effect on a vital target gene in Medfly, we understand how the results obtained favor this hypothesis. As described in the discussion, there are examples of RNAi-to-RNAi also in other species distant (e.g. L. decemlineata) and close (e.g. B. tryoni) to Ceratitis capitata that confirm how this method is potentially universal and versatile, and that can give us new ideas for the future. In the “conclusions” paragraph, in fact, we hope for the possibility and will to test this interesting method by targeting new vital genes not only in Medfly, but also in other close species (e.g. Bactrocera dorsalis). We have already identified potential new gene targets based on several works from the literature: Putative COPI coatomer, Chitin synthase II, and other subunits of the v-ATPase gene. We added this comment in the conclusions, for future investigation.
- Title: delete the last “adults
Done.
- Abstract:
1) the author said that “…which is poorly explored in dipteran species” in line 34. As I known, feeding dsRNA for RNAi have been explored in some labs. However, there was very few reports mentioning it because of its low effectiveness. Therefore I suggest to replace it by "…which is poorly reported in dipteran species".
Done.
2) in lines 40-41, the author said that “In contrast, we observed 79% mortality over seven days, which was associated with a decrease in mRNA levels of the three targeted genes”; I wonder what the treatment is; feeding dsRNA mixture of three genes?
Thanks for the notice. We have updated the text to clarify that this mortality is associated with treatment involving the administration of dsRNA against all three target genes.
- Introduction:
1) the introduction could be more concise and fucus on “simultaneous RNAi”, “v-ATPase”, and “gut nucleases”. For example, in the second paragraph, the author listed a lot of RNAi effects on various target genes, which were not necessary. While in the six paragraph, few information of “gut nucleases” was showed.
Thanks for the comment. We have eliminated the superfluous part of paragraph two and added to paragraph six a new and recent example of simultaneous RNAi against a vital gene target and a gut nuclease.
2) in line 80, the author mentioned the “systemic RNAi” in B. dorsalis; however, I could not find any evidence (even from the ref 14 cited by author) to support it.
Thanks for the comment. Given the reorganization of paragraph 2 and the purpose of this work, we have decided to eliminate this part regarding systemic RNAi and the related reference. Just for clarification, in Joga et al., 2016 review (Frontiers in Physiology, Volume 7 | Article 553), there are references concerning nor robust but present systemic RNAi in B. dorsalis (Chen et al., 2008; Li et al., 2015c; Zheng et al., 2015).
- Results:
1) as usual, the subsection titles of the “Results” should show the major related result of each part to facilitate reading, rather than introduce the research contents, please improve it.
Thanks for the suggestion. We have included titles that better introduce the underlying results.
2) the detailed method description and related information should not be shown in “Results”; this can facilitate the reader focusing on the major results; for example, in lines 322-332, the author introduced the detailed information of housekeeping gene selection and dsRNAs feeding experiments, which is better to be combined into “Materials and Methods”
Thanks for the comment. We have moved all information regarding the methodology used to the "Materials and Methods" section.
- Result “3.1”: As usual, the phylogenetic analysis was used to show the orthologous relationship of specific gene(s) to confirm the annotation of the gene sequence(s). I suggest the author add it
Thanks for the suggestion. We have added two phylogenetic trees (one for the vha68-1 gene and the other for dsRNases) to confirm and strengthen the analyses performed.
- Discussion: this part is lengthy and not very readable; for example, paragraphs 7-12 (lines 519-566) just concluded the results of RNAi effects on mRNA levels and mortalities and compared to previous related reports, which is better to be combined into one paragraph with more concise description.
Thanks for the comment. We have reorganized the discussion and added a table summarizing the works cited in the literature to make it easier to read.
- Reference: The format of the text must be unified. In addition, the species scientific names should be italicized (such as in ref 13)
Thanks for the comment. We have corrected the references as requested.
- Supplementary materials:
1) the supplementary tables could not be found
Thanks for the notice. The word version probably had some problems loading content. We have uploaded a pdf version.
2) figure S1 – S3 could not be found, and the basic information of each lane in each figure of electrophoresis gel should be added
Thanks for the notice. We met the requests.

Reviewer 2 Report
Comments and Suggestions for Authors
In the manuscript, a protocol for co-feeding of dsRNAs is described that target both an essential gene and two dsRNAse genes expressed in the gut to control medfly adults. The results are convincing but the quality of the manuscript needs to be approved to merit publication.
1) The authors should carry out phylogenetic analysis to illustrate the conservation of the proteins/genes that are targeted.
2) Semi-quantitative PCR: the number of cycles (Fig. 2) needs to be indicated.
3) It should be clearly described how much dsRNA was used to target each dsRNAse in the „dsRNAses“ condition or to target the individual genes in the „Mixed“ condition.
4) For the quantification of silencing, it should be clearly stated how many biological repeats were carried out. In addition, the authors should clearly indicate what constitutes a repeat (group of 4 flies or single flies).
5) In the mortality experiment, the authors should indicate how many biological repeats were used at each time point (legend of figure 3). In Materials and Methods, this experiment is not described in detail („dsRNA feeding“ refers only to the silencing experiments).
6) The text of the introduction and discussion needs to be revised. Experiments from the literature are discussed in detail without much organization. The authors should provide a Table that summarizes the experiments from the literature that are relevant and indicate in the Table the issues that they consider important. The introduction needs to be shortened considerably with respect to elements that re-appear in the discussion.
Comments on the Quality of English LanguageEnglish language is fine. Some minor corrections may be necessary, e.g. lines 82-83 and 112-113.
Author Response
1) The authors should carry out phylogenetic analysis to illustrate the conservation of the proteins/genes that are targeted.
Thanks for the suggestion. We added the out phylogenetic analysis of the three proteins involved in our study.
2) Semi-quantitative PCR: the number of cycles (Fig. 2) needs to be indicated.
Thanks for the comment. We added the number of PCR cycles in paragraph 2.4. (Materials and methods).
3) It should be clearly described how much dsRNA was used to target each dsRNAse in the „dsRNAses“ condition or to target the individual genes in the „Mixed“ condition.
Thanks for the comment. Although this information is present in Table S3 of the supplementary data, we have added the amounts of dsRNA used for each treatment in paragraph 2.5. (materials and methods).
4) For the quantification of silencing, it should be clearly stated how many biological repeats were carried out. In addition, the authors should clearly indicate what constitutes a repeat (group of 4 flies or single flies).
Thanks for the comment. For each treatment we always used 4 biological replicates. We consider the single fly as a biological replicate. We have made this information clearer in paragraph 2.6. (Materials and methods).
5) In the mortality experiment, the authors should indicate how many biological repeats were used at each time point (legend of figure 3). In Materials and Methods, this experiment is not described in detail („dsRNA feeding“ refers only to the silencing experiments).
Thanks for the comment. We have added this information to the legend of figure 3 as requested. Additionally, we have added a more detailed description of the experiment in section 2.5. (Materials and methods).
6) The text of the introduction and discussion needs to be revised. Experiments from the literature are discussed in detail without much organization. The authors should provide a Table that summarizes the experiments from the literature that are relevant and indicate in the Table the issues that they consider important. The introduction needs to be shortened considerably with respect to elements that re-appear in the discussion.
Thanks for the comment. We understand these critical points, and we have tried to meet the reviewer request. We shortened the introduction and added Table 1 with the references.

Round 2
Reviewer 1 Report
Comments and Suggestions for Authors
I think this revised version of manuscript has met the requirements for publication and recommend its acceptance.
Reviewer 2 Report
Comments and Suggestions for Authors
The manuscript was revised appropriately and can be accepted for publication.